# Rapid Prototyping of Multi-Functional and Biocompatible Parafilm^®^-Based Microfluidic Devices by Laser Ablation and Thermal Bonding

**DOI:** 10.3390/mi14030656

**Published:** 2023-03-14

**Authors:** Yuanyuan Wei, Tianle Wang, Yuye Wang, Shuwen Zeng, Yi-Ping Ho, Ho-Pui Ho

**Affiliations:** 1Department of Biomedical Engineering, The Chinese University of Hong Kong, Shatin, Hong Kong 999077, China; 2Bionic Sensing and Intelligence Center, Institute of Biomedical and Health Engineering, Shenzhen Institute of Advanced Technology, Chinese Academy of Sciences, Shenzhen 518055, China; 3XLIM Research Institute, UMR 7252, University of Limoges, 123 Avenue Albert Thomas, 87060 Limoges, France; 4Light, Nanomaterials & Nanotechnologies (L2n), CNRS-ERL 7004, Université de Technologie de Troyes, 10000 Troyes, France; 5Centre for Biomaterials, The Chinese University of Hong Kong, Hong Kong 999077, China; 6Hong Kong Branch of CAS Center for Excellence in Animal Evolution and Genetics, Hong Kong 999077, China; 7The Ministry of Education Key Laboratory of Regeneration Medicine, Hong Kong 999077, China

**Keywords:** microfluidics, laser ablation, thermal bonding

## Abstract

In this paper, we report a simple, rapid, low-cost, biocompatible, and detachable microfluidic chip fabrication method for customized designs based on Parafilm^®^. Here, Parafilm^®^ works as both a bonding agent and a functional membrane. Its high ultimate tensile stress (3.94 MPa) allows the demonstration of high-performance actuators such as microvalves and micropumps. By laser ablation and the one-step bonding of multiple layers, 3D structured microfluidic chips were successfully fabricated within 2 h. The consumption time of this method (~2 h) was 12 times less than conventional photolithography (~24 h). Moreover, the shear stress of the PMMA–Parafilm^®^–PMMA specimens (0.24 MPa) was 2.13 times higher than that of the PDMS–PDMS specimens (0.08 MPa), and 0.56 times higher than that of the PDMS–Glass specimens (0.16 MPa), showing better stability and reliability. In this method, multiple easily accessible materials such as polymethylmethacrylate (PMMA), PVC, and glass slides were demonstrated and well-incorporated as our substrates. Practical actuation devices that required high bonding strength including microvalves and micropumps were fabricated by this method with high performance. Moreover, the biocompatibility of the Parafilm^®^-based microfluidic devices was validated through a seven-day *E. coli* cultivation. This reported fabrication scheme will provide a versatile platform for biochemical applications and point-of-care diagnostics.

## 1. Introduction

Microfluidics is a rapidly growing research discipline due to its highly attractive features. Its advantages include device miniaturization, a drastic reduction in reagent consumption, portability, low cost, ease of volume production, and compatibility with conventional integrated circuit manufacturing processes. Microfluidic devices have found applications in chemical analysis, the processing and analysis of biological samples, and disease diagnostics [1,2]. Many researchers have utilized flexible polymers such as polydimethylsiloxane (PDMS) as the substrate, where patterned structures could be imprinted and transferred from a mold [3,4].

PDMS has played an important role in microfluidics because of its properties such as low surface interfacial free energy, optical transparency, and gas permeability. Besides, PDMS has been used for built-in functional actuation structures such as micropumps and microvalves due to its large elasticity [5,6,7]. However, recent studies have revealed some drawbacks of PDMS including short-term stability after surface treatment (such as hydrophilic stability after oxygen plasma treatment), the adsorption of hydrophobic molecules, swelling in organic solvents, and incompatibility with high-pressure operations [5,8].

Compared to PDMS, thermoplastic materials present better mechanical properties and robustness [9]. Especially, PMMA (poly (methyl methacrylate), commonly pronounced as acrylic) is a non-porous solid that has excellent optical transparency and chemical inertness. Among different fabrication methods such as microinjection molding, hot embossing, casting, and reactive ion etching, CO_2_ laser micromachining offers high versatility and cost-effective manufacturing [10].

Besides patterning microfluidic chips, the effective and stable bonding of thermoplastics is also a barrier to rapid prototyping and high-throughput manufacturing [11]. Among solvent bonding, adhesive bonding, surface activation treatments (e.g., Plasma), and microwave bonding, thermal bonding is commonly adopted due to its merits of accessibility and simplicity [12]. Considering the advantage of simple and uniform surface bonding, the bonding strength of thermal bonding increases by increasing the bonding temperature and pressure [13]. However, the use of bulky press machines and heaters may distort the device geometry [14]. The low-temperature bonding property of Parafilm^®^ (thickness: ~0.13 mm; melting point: 60.0 °C) offers a solution to this problem.

Parafilm^®^ is well-known as a universal laboratory staple with the properties of being colorless, odorless, and highly waterproof. Utilizing Parafilm^®^ as a low-temperature bonding material to approve rapid prototyping methods with multiple materials has been successfully applied to fabricate paper-based analytical devices (μPADs) in a non-cleanroom manner. These fast prototyping methods can be classified as the mask-guided infilling of a sheet of Parafilm^®^ [15], direct infilling of photolithographically patterned Parafilm^®^ [16], craft-cut Parafilm^®^ into porous paper using a heated press [17], cutter and thermal bonding [18], and laminated and infused Parafilm^®^ into l-paper (partial fusion) and i-paper (complete fusion) [19]. However, all of the previous works are limited to paper-based substrates, which possess poor mechanical strength. Additionally, none of the works has measured and quantified the bonding strength of fabricated chips, which has led to the absence of a direct comparison of the reliability of the Parafilm^®^-based chips and PDMS-based chips. Moreover, the elasticity of Parafilm^®^ provides the possibility of building up functional actuation microfluidic devices, which has never been demonstrated. Especially, for [15], it cannot produce separated channels with well-defined dimensions because each channel needs an independent shadow mask. For [17], the special resolution is limited due to the lateral spreading of paraffin wax together with the inward moving during the heated press.

In this paper, we report a rapid and low-cost prototyping method for the fabrication of microfluidics devices. This multi-functional and biocompatible Parafilm^®^-based analytical microfluidic device prototyping method consists of laser ablation and thermal bonding (PbLT). A schematic of this technique is shown in Figure 1. Parafilm^®^ and PMMA materials were shaped into desired patterns by laser ablation. Then, they were effectively bonded together by thermal fusion to form a 3D structured microfluidic chip. Here, Parafilm^®^ served as both the bonding agent and the functional membrane.

We studied and optimized the following parameters: the resolution of the microchannel, channel deformation degree, biochip leakage degree, bonding strength, and surface roughness of the cross-section. We also validated the performance of the fabricated devices through the following tests: (a) test of the mechanical properties of Parafilm^®^-based elastomer; (b) measurement of the bonding strength across different substrates and incubation conditions; (c) demonstration of actuators including microvalves and micropumps; (d) investigation of the biocompatibility of the PbLT-fabricated bioreactor. In the end, we made a comparison between our PbLT approach and other fabrication techniques in terms of major application attributes. Our method showed greater superiority in terms of set-up cost, cycle time, 3D capability, and throughput. We can assert that the reported PbLT fabrication approach offers an economical and practical alternative for the rapid prototyping of robust and complex microfluidic chips.

## 2. Materials and Methods

### 2.1. Laser Ablation

Parafilm^®^ M (Pechiney Plastic Packaging Company Milwaukee, WI, USA) and raw-cast transparent PMMA sheets in various thicknesses (1–5 mm) were used in our experiments. All laser-cutting procedures were performed using a commercial CO_2_ laser system (Maximum power: 80.0 W, focal spot size: ~0.25 mm, Model CMA960, GD Han’s Yueming Laser Group Co., Ltd., Hong Kong, China) as shown in Appendix A. All microfluidic chip designs were implemented using the software Solidworks2016 and AutoCAD2018. The designed patterns were saved as DXF (Drawing Interchange Format) files. Software SmartCarve4 converts the DXF files into a series of commands for adjusting the laser fabrication process.

Three crucial parameters affecting the profiles of the ablated channels were adjusted during the fabrication process [20]. They were focusing distance (measured by taking the actual distance between the laser head and the top of the sample) (*k*), laser output power (*pwr*), and laser head speed (*v*). Briefly, *k* was fixed at 6.0 mm, *pwr* was set between 4.0 and 80.0 W, and *v* was in the range of 0 mm/s to 100 mm/s. A series of scribing trials on Parafilm^®^ and PMMA were carried out accordingly. For data presented in this study, five independent ablation processes with identical settings were performed.

### 2.2. Thermal Bonding

After laser ablation, Parafilm^®^ and PMMA slides were aligned and assembled manually. The assembly was then placed on a hotplate (MS-280-H, Hangzhou Jingfei Instrument Technology Co., Ltd., Hangzhou, China). Different calibrated static pressures were added to the top utilizing heavy loads (e.g., stone block). The bonding temperature was varied between 25.0 and 85.0 °C. The 5.0 kg stone block was capable of introducing a bonding pressure between 8.3 and 33.3 kPa. For uniform pressure distribution, a 5 mm-thick PMMA spacer was placed between the block and the specimen (Figure 1). The thermal bonding process included (a) heating for 20 min and (b) cooling to room temperature over 40 min. The static pressure was applied continuously.

### 2.3. Characterization by Microscopy

The geometry of ablated microchannels was observed by an optical microscope (Olympus BH2-UMA, Olympus Corporation, Tokyo, Japan) equipped with a digital camera (PROMICRA, Evropska, Czech Republic). Trypan Blue water (Yik Fung Scientific Co., Hong Kong, China) was injected into the microchannels for better visualization. Channel widths were measured across the microchannel by the software ImageJ. Hydrophobic treatment was carried out utilizing a water-repellent agent (47100, Aquapel, Pittsburgh, PA, USA). After injecting Aquapel into the microchannels, the specimen was baked at 50.0 °C on a hotplate until Aquapel dried out. To obtain a cross-sectional structure, the PbLT-fabricated chip was sliced utilizing the same CO_2_ laser system. The cross-sectional structure of microchannels was examined by scanning electron microscopy (SEM) (FEI QUANTA 400F, Thermo Scientific, Waltham, MA, USA). Before SEM imaging, the specimen was sputter-coated with gold. Imaging and monitoring of bacteria cultivation were carried out with a fluorescence microscope (Eclipse Ti-U, Nikon, Tokyo, Japan).

### 2.4. Mechanical Characterization

Mechanical characterization was conducted using a tensile tester (QT/1L, QTest^TM^, MTS Systems Corporation, Eden Prairie, MN, USA). Tensile testing was conducted to investigate the performance of Parafilm^®^ when applying normal and extreme forces. Before tensile testing, Parafilm^®^ specimens were applied to different heating treatments (room temperature, 50.0 °C, and 85.0 °C) and static pressure treatments (0 and 0.4 kPa). The results indicated the mechanical property changes of Parafilm^®^ after the thermal bonding process. Parafilm^®^ specimens with different gauge widths (10.0 mm, 15.0 mm, 20.0 mm) were prepared utilizing the same CO_2_ laser system. The grip section had an identical width of 25.0 mm. The gauge length was 9.0 mm. Specimens were held in grips with a gauge length of 90.0 mm and stretched at a constant strain rate of 1.0% per minute.

The shear stress was examined by shear tests. Shear tests were carried out using the same tensile tester (shear mode settings). Specimens were fabricated with the PbLT fabrication process utilizing different substrate materials (PMMA, PVC, glass). For each specimen, the Parafilm^®^ slide was sandwiched between two identical mismatched substrates. The Parafilm^®^ slides were varied as 10.0 mm × 10.0 mm, 20.0 mm × 10.0 mm, and 40.0 mm × 10.0 mm.

Different heating temperatures (50.0 °C and 85.0 °C) were applied during thermal bonding. A 10.0 mm-length gauge pressed one of the substrates with a constant crosshead speed of 1.0 mm/min. The maximum force the bonding structure could resist was recorded as its peak load.

All testing experiments were carried out at room temperature. Each experiment was repeated five times.

### 2.5. Microvalve and Micropump

A custom-made valve controller system was used for actuating the PbLT-fabricated microvalves and micropumps. The system consists of an ELVEFLOW microfluidic flow controller (OB1 MK3+ Microfluidic Flow Controller, ELVEFLOW, Paris, France) and ELVEFLOW Smart Interface software. This system is capable of controlling compressed nitrogen gas at a maximum pressure of 0.2 MPa. The pressurized nitrogen gas from the ELVEFLOW controller was injected into the PbLT-fabricated chips for actuation experiments.

### 2.6. Bacteria Cultivation in Bioreactor

The frozen stock of recombinant Escherichia coli (*E. coli*) expressing green fluorescent protein (GFP) was resuspended in Lysogeny broth (25 g/L, Merck Limited, Darmstadt, Germany) supplemented with 50 µg/mL of kanamycin (Merck Limited, Darmstadt, Germany), transferred onto an agar plate, and cultivated at 37.0 °C for 2 days. Populations were established by inoculating 5 mL of a liquid medium with a single isolated colony and cultivated at 37.0 °C for another day. The cultivated bacteria were diluted to 1 × 10^6^ cells/mL for the subsequent experiments. Before the loading, the bioreactor was sterilized by rinsing with 75.0% ethanol followed by deionized water, air-drying, and irradiation with ultraviolet light for 30 min. Next, 100 µL of bacteria suspension were manually injected into each chip. The inlet and outlet were then blocked to prevent evaporation and contamination during the 7-day cultivation at 37.0 °C.

## 3. Results and Discussion

### 3.1. Effects of Laser Power and Laser Scanning Speed

The laser ablation process is shown in Figure 2a. In Figure 2b, the laser power increased from 4.0 W to 8.0 W, and the microchannels on the Parafilm^®^ surface became continuous. So as to achieve a minimum continuous channel with a minimum standard deviation (249.31 ± 3.7 μm) the laser power was set to 12.0 W (Figure 2c). It was found that the laser head speed affected the channel width and the channel uniformity. As shown in Figure 2d, when the laser head speed increased, the channel width decreased. The larger the standard deviation, the lower the channel uniformity. So as to achieve a minimum uniform channel (249.79 ± 7.66 μm) the laser head speed was set to 60 mm/s.

Similar results were carried out for the laser ablation process of the 1 mm PMMA substrate as Figure 2e,f shows. The minimum channel width of 171.45 ± 4.8 μm was achieved when the laser power was 36.0 W and the laser head speed was 60 mm/s.

For the subsequent investigations, the laser output power and laser head speed were set as *pwr* = 12.0 W and *v* = 60 mm/s for the Parafilm^®^, and *pwr* = 36.0 W and *v* = 60 mm/s for the 1 mm PMMA substrate. Correspondingly, the optimized parameters for the 2 mm PMMA substrate were *pwr* = 80.0 W and *v* = 60 mm/s; for the 2 mm PVC substrate, *pwr* = 80.0 W and *v* = 60 mm/s for 2 repetitions; for the 2 mm glass substrate, *pwr* = 80.0 W and *v* = 10 mm/s for 80 repetitions (optimization process not shown here). All the chips, microvalves, micropumps, and bioreactors reported in this study were fabricated using this optimized laser ablation procedure.

### 3.2. Effects of Bonding Temperature and Static Pressure on Parafilm^®^ Deformation

In the tensile testing, the ultimate tensile stress of Parafilm^®^ (3.49–3.94 MPa) was 3.43 times higher than that of the PDMS specimens (0.89 MPa, base-to-curing ratio of 10:1). Compared to the ultimate tensile stress of PDMS from reference [21] (3.51–7.65 MPa), Parafilm^®^ has a comparable tensile stress. From Figure 3a, the ultimate tensile stress of the Parafilm^®^ specimens (gauge width: 15.0 mm) decreased when the heating temperature increased. The same results were observed for the Parafilm^®^ specimens with widths of 10.0 mm and 20.0 mm (Appendix A). The higher ultimate tensile stress caused by adding static pressure during heating is shown in Appendix A. Compared to adding no pressure, the one to which 0.4 kPa static pressure was applied had a higher ultimate tensile stress, indicating better elasticity. The above results show the capability of Parafilm^®^ to perform as a functional membrane. The information also indicates the mechanical property changes of Parafilm^®^ after the thermal bonding process.

In the experimental study of deformation and leakage, the Parafilm^®^ layer was designed for five channels (channel width: 500 μm; channel length: 3 cm). The ablated Parafilm^®^ layer was sandwiched between two 2 mm PMMA layers. Correspondingly, five paired inlets and outlets were ablated on the top-layer PMMA sheet (Figure 3b).

The thermal bonding was carried out under the conditions with different combinations of temperatures (25.0 °C, 50.0 °C, and 85.0 °C) and static pressures (low pressure: 8.3 kPa, middle pressure: 16.7 kPa, high pressure: 33.3 kPa). After injecting Trypan Blue aqueous solution into the chips, the channel width and channel length of the Parafilm^®^ layer for each chip were measured.

The deformation of the fabricated chip was characterized by calculating as follows:(1)$${∆}_{width}=\frac{W_{before\;bonding}-W_{after\;bonding}}{W_{before\;bonding}}\times 100\%$$
where ${∆}_{width}$ is the deformation degree of the channel, $W_{before\;bonding}$ and $W_{after\;bonding}$ are the channel widths before and after the thermal bonding process, respectively.

From Figure 3c, the higher the bonding temperature, the larger the deformation degree of the fabricated Parafilm^®^ channel. Overall, the deformation degrees of all three different temperatures were less than 15%. The deformation degree of the channel edges was larger than that in the middle part of the channel.

Similarly, the leakage degree was defined as:(2)$${∆}_{leakage}=\frac{L_{leakage}}{L_{ablated\;channel}}\times 100\%$$

Here, ${∆}_{leakage}$ is the leakage degree of the channel. $L_{leakage}$ is the length of the channel from which Trypan Blue water leaked out. $L_{ablated\;channel}$ is the measured channel length of the ablated channel.

In Figure 3d, for the chips bonded at 25.0 °C (room temperature), obvious leakage was observed within 10 min after injecting Trypan Blue water. As expected, without the assistance of heating, the adhesion force between the Parafilm^®^ and PMMA sheets was not strong enough to prevent leakage. Minimal leakage was observed for the chips fabricated at 50.0 °C. No leakage was observed for the chips fabricated at 85.0 °C. The fabrication static pressure was 33.3 kPa for all three heating temperatures.

Overall, compared to PDMS, Parafilm^®^ had a comparable resistance to tensile stress. The ultimate tensile stress decreased when Parafilm^®^ was heated. The higher the heating temperature, the lower the ultimate tensile stress. When Parafilm^®^ was used as a bonding agent, the fabricated chips with a static pressure of 33.3 kPa and 85.0 °C heating temperature had no leakage. The deformation degree was less than 15.0%. Minimal leakage was observed for the chips fabricated at 50.0 °C. When we used Parafilm^®^ as a bonding agent, the bonding temperature was kept at 85.0 °C. For membrane microfluidic structures, the heat-assisted bonding process was carried out at 50.0 °C. All chips, microvalves, micropumps, and bioreactors reported in this study were prepared using this optimized bonding procedure.

### 3.3. Mechanical Strength of Parafilm^®^ Bonding

Here, we characterized the bonding strength of fabricated chips by shear tests and gas leakage tests. Following the optimized PbLT process, the fabricated chips of different bonding areas (100 mm^2^, 200 mm^2,^ and 400 mm^2^) were tested.

From Figure 4a, the fabricated chips with a bonding temperature of 85.0 °C had a larger resisted shear force than those of 50.0 °C. The larger the bonding area, the larger the resisted shear force. The higher the shear force, the higher the bonding strength, and the higher the resistance to leakage and high-pressure applications. It was noted that the values of all of the 85.0 °C PbLT-fabricated chips (0.24 MPa for 1 cm^2^) were 2.17-fold higher than those of the PDMS–PDMS specimens (0.08 MPa) and 0.58-fold higher than those of the PDMS–Glass specimens (0.16 MPa) by oxygen plasma-assisted bonding [22].

Compressed nitrogen gas (maximum 0.2 MPa) was injected into the fabricated chips for gas leakage tests. For the chips fabricated at 85.0 °C, minimal leakage was observed. Moreover, complete Parafilm^®^ remained on the substrate surface after bursting out, indicating a high bonding strength and tight sealing ability. For the chips fabricated at 50.0 °C, gas began to leak at 0.06 MPa, with all Parafilm^®^ removed from the substrate (Appendix A).

The resistance of the fabricated chips to the cell culture conditions was studied as shown in Figure 5b. It also provides a reference for evaluating the long-term mechanical stability of the biochips. The fabricated biochips (bonding area: 200 mm^2^) were placed in an incubator (37.0 °C and 5.0% CO_2_ atmosphere) for seven days. After incubation, an evident decrease in bonding strength (45.9%) was observed for the biochips fabricated at 85.0 °C. Similar results were observed for the bonding of 100 mm^2^ and 400 mm^2^ biochips (Appendix A). However, there was a slight increase for the biochips fabricated at 50.0 °C. Presumably, this resulted from the effects of the humid conditions on the Parafilm^®^ bonding layer.

The same shear test experiments were carried out for chips fabricated by accommodating several additional substrate materials. From Figure 4c, the bonding strength of PVC substrates and glass substrates (bonding area: 200 mm^2^) was comparable to that of PMMA. Additionally, chips fabricated at 85.0 °C had larger bonding strengths than the ones at 50.0 °C. Similar results were observed for the bonding of 200 mm^2^ and 400 mm^2^ chips (Appendix A).

### 3.4. Micropump and Microvalve Characterization

Besides working as a bonding agent, Parafilm^®^ can also be used as a functional membrane to demonstrate practical actuation devices. In Figure 5a, Parafilm^®^-based normally open microvalves were demonstrated to investigate their functions in impeding aqueous liquid flow. Pressurized nitrogen gas was injected from the inlet to actuate the Parafilm^®^ membranes, leading to a Parafilm^®^ deformation for actuating the liquid flow. A peristaltic micropump was developed by the integration of three interconnected microvalves on a common liquid channel.

As shown in Figure 5b,c, the micropump was constructed of a control layer and a liquid layer. The received Parafilm^®^ was directly used for fabricating the type 1 micropump. Stretched Parafilm^®^ was used as an optimized type 2 micropump. The effect of the actuation gas pressure and the frequency of actuation on the pumping flow rate were studied, encompassing pressures of 25, 50, and 200 kPa and frequencies of 1, 2, 3, 5, and 20 Hz. The flow rate was measured by the volume of injected Trypan Blue water filled in the tubing (I.D. 1.6 mm, AVX42003, Saint-Gobain Performance Plastics, Paris, France) during a specific period. The bonding strength of the thermally bonded PMMA–Parafilm^®^–PMMA layers constituting the microvalve architecture was tested. No burst failure was observed for gas pressures of up to 200 kPa.

To the best of our knowledge, this is the first attempt to demonstrate a microvalve and micropump utilizing original Parafilm^®^ (type 1 micropump) and stretched Parafilm^®^ (type 2 micropump), respectively. The reported micropumps possess a high flow rate (>20 μL/s) compared to conventional PDMS-based micropumps, which have maxima of 0.12 μL/s [9] and 0.10 μL/s [23], respectively. The type 2 micropump overcame the short-duration limitation of the type 1 micropump by utilizing stretched Parafilm^®^. It has been observed that for the type 1 micropump, the flow rate decreased from over 20.0 μL/s to 0 μL/s within 25 s, which was presumably due to the fatigue failure of the Parafilm^®^ (Appendix A). However, the pumping performance of the type 2 micropump was relatively more stable and durable, as shown in Figure 5d. For the type 2 micropump, the performance related to actuation gas pressure and gas source frequency is shown in Appendix A.

### 3.5. SEM Characterization of Microchannels

The SEM images of the PbLT-fabricated chip cross-sections are shown in Figure 6 to present the surface profile. In Figure 6a, the bonding temperature is 85.0 °C and the static pressure is 33.3 kPa. Two boundary lines can be observed between the layers, showing the interface between the PMMA substrate and Parafilm^®^. After hydrophobic treatment of the microchannels, as Figure 6b shows, the surface of the microchannel became smoother. The boundaries between the PMMA and Parafilm^®^ became blurred. 

To verify whether non-specific binding was presented, 1 μm and 10 μm polystyrene (PS) beads (Sigma-Aldrich Inc., Seoul, Republic of Korea) were introduced into the microchannels. In Figure 6c, a large number of PS beads were concentrated at the interface of the microchannel regions. Obvious PS bead clusters were observed at the corner regions. Presumably, this resulted from the rough surface of the melted Parafilm^®^ and PMMA, which was caused by unequal heat distribution during the fabrication process [20]. After hydrophobic treatment, PS beads presented a more scattered distribution as shown in Figure 6d. The SEM images revealed that the surface roughness was largely improved.

### 3.6. Bioreactor Characterization

GFP-expressing *E. coli* were cultivated in the PbLT-fabricated bioreactor, as shown in Figure 7a, to assess biocompatibility. As observed from the fluorescent signal shown in Figure 7b, the bacteria underwent proliferation and doubled roughly every 30 min, which is a similar doubling time to that typically observed in bulk liquid culture [24]. At one day post-cultivation, the growth of *E. coli* was observed to have arrested, presumably arriving at the stationary phase or non-growth phase. When nutrients were consumed at three days and seven days post-cultivation in a confined space, the green fluorescence was significantly decreased owing to the death of *E. coli*. (Appendix A). After resuspension in the same PbLT-fabricated reactor with buffer, the bacteria reproliferated. The observed phenomena are similar to *E. coli* cultivation in typical liquid culture [25]. In this cultivation experiment, 3 out of 15 PbLT-fabricated bioreactors leaked soon after injecting the medium (Appendix A). No leakage was observed for the remaining bioreactors after the seven-day cultivation.

### 3.7. Comparison of the PbLT Method and Other Fabrication Methods

Conventional microfluidic chip fabrication methods, such as soft lithography and injection molding, are often very costly and time-consuming. The long cycle time and the necessity for expensive facilities are the barriers to rapid prototyping. This limits the development of microfluidics, especially during the early stage of research. Poisonous chemicals (such as HF and SU8 solvent) are also contrary to the original goals of microfluidics development: inexpensiveness, convenient, portability, environmentally friendly, and disposability [26]. The prototyping method of our work provides a low-cost and time-saving solution to the fabrication of microfluidic chips, as Table 1 shows. A typical several-day fabrication process can be optimized within 2 h, with easily obtained fabrication materials (Parafilm^®^, PMMA, PVC, and glass).

## 4. Conclusions

Microfluidic device fabrication is an actively developing field. Despite the advantages of conventional soft lithography utilizing PDMS, expensive facilities and time-consuming methods are still significant challenges. The alternatives are to replace the facilities or to replace the materials (including substrates and bonding agents). In this paper, we specifically addressed the limitations of conventional soft lithography and PDMS material. We developed a rapid prototyping method, allowing the fabrication of customized multi-functional microfluidic chips from Parafilm^®^ within 2 h. The laser ablation process and one-step thermal bonding process were optimized to achieve high a bonding strength and non-leakage performance. The resolution was limited by the laser head, which can be improved by changing the laser head to a smaller spot area [27]. The mechanical properties, surface characterization, actuation performances, and biocompatibility of fabricated devices were studied.

Here, Parafilm^®^ works as both a bonding agent and a functional membrane. The high ultimate tensile stress of Parafilm^®^ was tested, allowing for the demonstration of high-performance actuators such as microvalves and micropumps. Multiple easily accessible materials such as polymethylmethacrylate (PMMA), PVC, and glass slides were applied as substrates. The cross-section surfaces of fabricated chips were characterized by SEM, and the surface roughness was improved by hydrophobic treatment. The bonding strength was characterized using a tensile machine and leakage tests. The study showed that the chips fabricated at 85.0 °C were able to achieve a limit of 0.24 MPa of shear stress and at least 0.2 MPa of gas burst pressure. A gas-actuated microvalve and micropump were developed and operated without leakage, with a pumping speed much greater than reported micropumps. The biocompatibility of fabricated bioreactors was validated by cultivating GFP-expressing *E. coli* for seven days.

This reported fabrication scheme provides a versatile platform for various biochemical applications. The easy obtainability of fabrication materials, as well as facilities, makes it a rapid and low-cost fabrication method. We envision that this rapid prototyping method will be readily adopted by biochemical laboratories and developed into point-of-care systems shortly.

## Figures and Tables

**Figure 1 micromachines-14-00656-f001:**
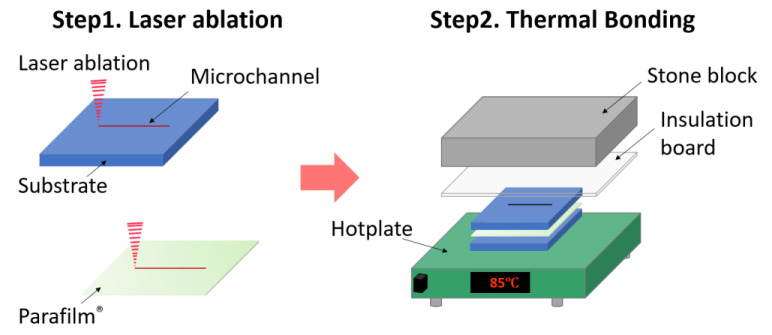
Fabrication process flow of the Parafilm^®^-based laser ablation and thermal bonding (PbLT) fabrication method. Parafilm^®^ and different substrates were cut to desired patterns (Step 1). Then, they were aligned and assembled manually. Followed by thermal bonding (Step 2), the assembly was placed on a hotplate for constant heating. An insulation board and stone block were superposed for uniform static pressure.

**Figure 2 micromachines-14-00656-f002:**
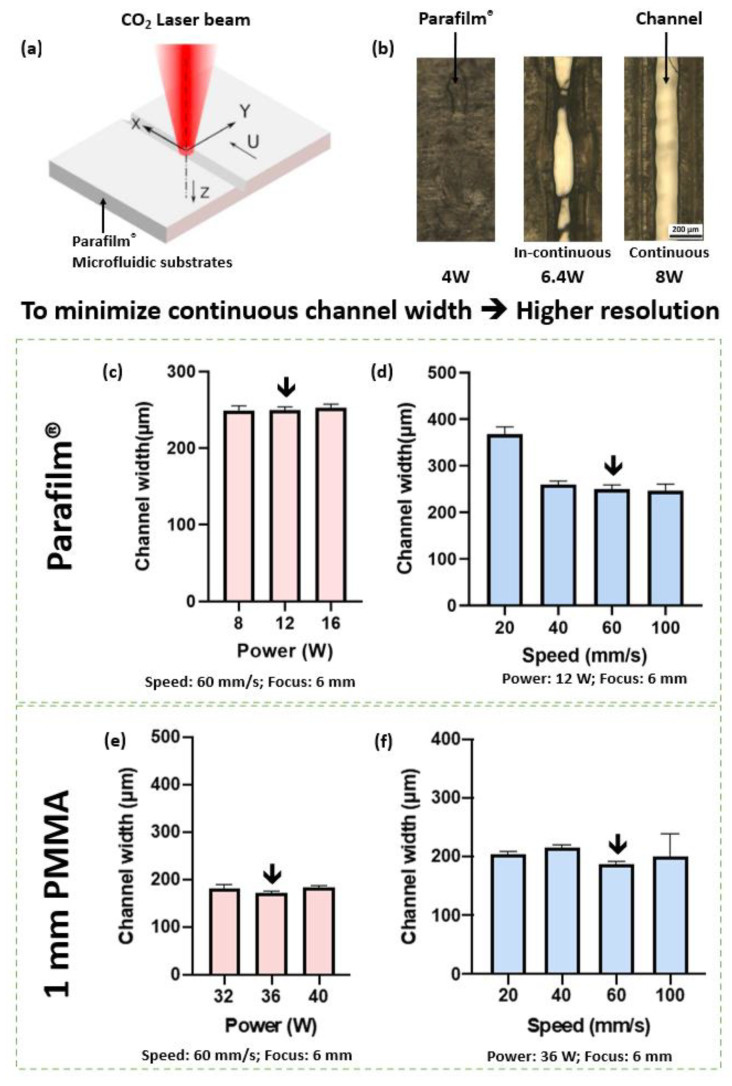
Illustration of laser ablation process (**a**) and effects of laser ablation parameters on the geometry of laser-plotted pattern (**b**–**f**). Optimized parameters were initiated. The optimization of laser output power and laser head speed was carried out with a focus of 6.0 mm. For each data point in figures (**c**–**f**), the experiment was carried out 5 times. (**a**) By laser ablation, Parafilm^®^ and different substrates were cut to desired patterns. (**b**) Laser output power was set to over 8.0 W to obtain a continuous channel when cutting Parafilm^®^. (**c**) Laser output power vs. channel width for cutting Parafilm^®^ with laser head speed of 60 mm/s. (**d**) Laser head speed vs. channel width for cutting Parafilm^®^ with laser output power of 12.0 W. (**e**) Laser power vs. channel width for cutting 1 mm PMMA with laser head speed of 60 mm/s. (**f**) Laser head moving speed vs. channel width for 1 mm PMMA with laser output power of 36.0 W.

**Figure 3 micromachines-14-00656-f003:**
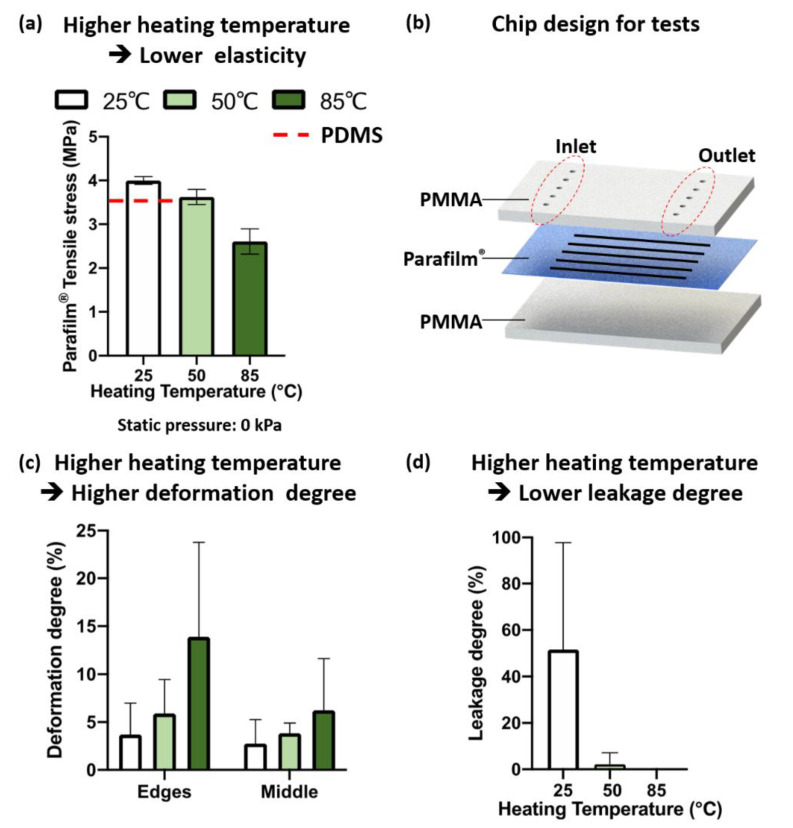
Effects of temperature and pressure on Parafilm^®^ mechanical properties and deformation. (**a**) The higher the heating temperature, the lower the ultimate tensile stress of Parafilm^®^. Width of Parafilm^®^ specimens: 15 mm. No pressure was applied to the Parafilm^®^ specimens. (**b**) Chip design for leakage and deformation tests. Deformation degree (**c**) and leakage degree (**d**) of Parafilm^®^ microchannels with different heating temperatures. Static press pressure: 33.3 kPa.

**Figure 4 micromachines-14-00656-f004:**
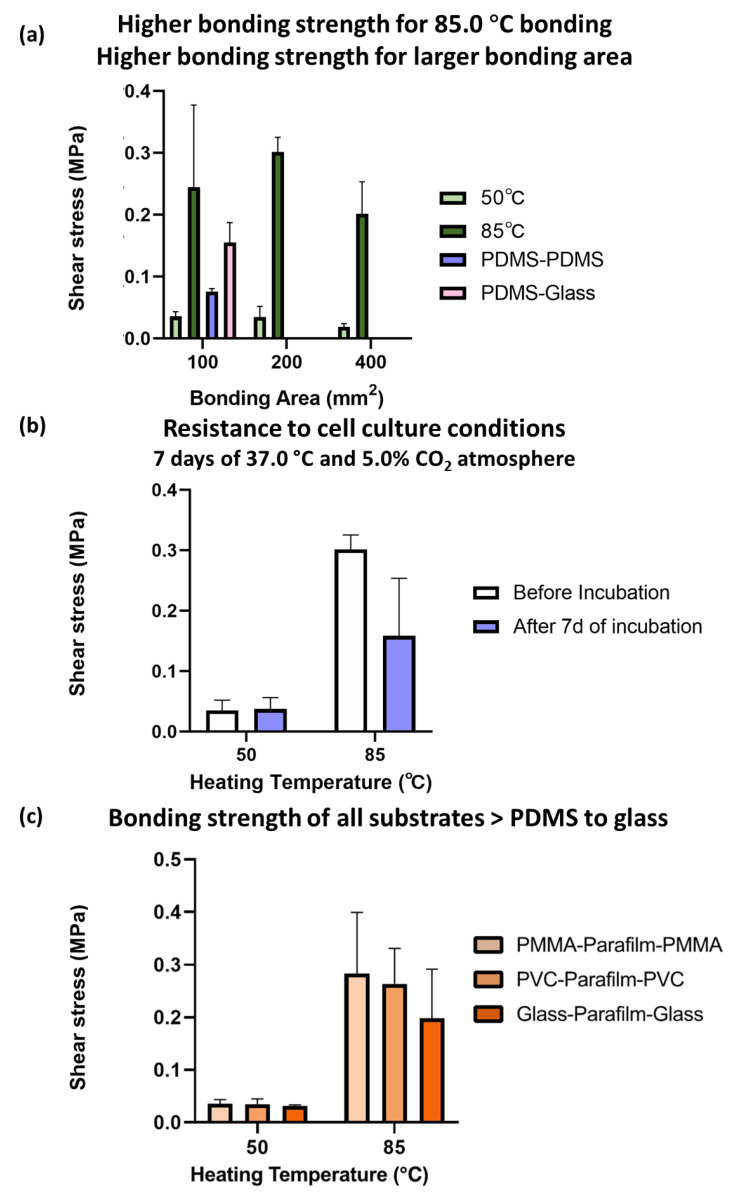
Shear stress for PbLT biochips with thermal bonding temperatures of 50.0 °C and 85.0 °C. The higher the shear force, the higher resistance to leakage and high-pressure applications. (**a**) Chips fabricated at thermal fusion temperatures of 85.0 °C had higher bonding strengths than those at 50.0 °C. The bonding areas were 100 mm^2^, 200 mm^2^, and 400 mm^2^, respectively. (**b**) Chips fabricated at thermal bonding temperatures of 85.0 °C had better resistance to cell culture conditions than those at 50.0 °C. The cell culture conditions were simulated by applying 7 days of 37.0 °C and 5.0% CO_2_ atmosphere. Bonding area: 200 mm^2^. (**c**) PbLT-fabricated chips with PMMA, PVC, and glass substrates had strong bonding strengths. Bonding area: 200 mm^2^.

**Figure 5 micromachines-14-00656-f005:**
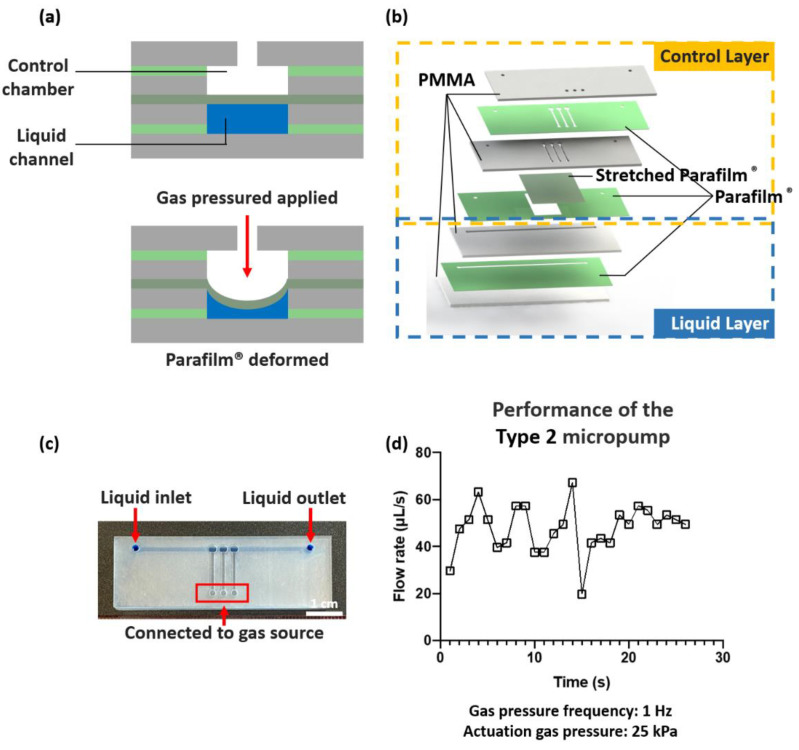
Fabrication and characterization of the normally open microvalve and micropump. (**a**) Operation design of the microvalve. When gas pressure was applied to the control chamber, the Parafilm^®^ was deformed to actuate liquid flow. (**b**) The 3D architecture of the fabricated micropump with multiple microvalves. Parafilm^®^ was utilized to work as both the bonding agent and functional membrane. (**c**) Photograph and schematic showing the fabricated micropump. The control chamber is on the top and the liquid channel is at the bottom. Scale bar = 1 cm. (**d**) Effect of actuation pressure and frequency on actuating flow rate.

**Figure 6 micromachines-14-00656-f006:**
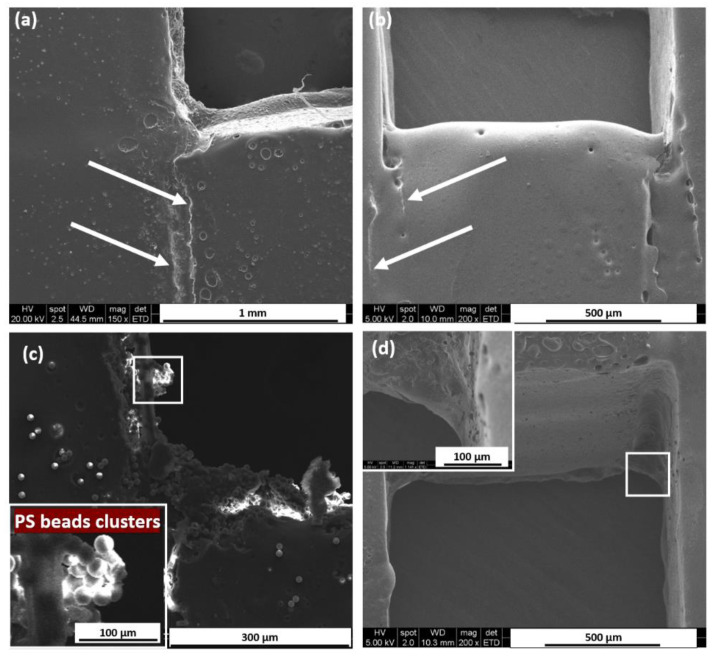
Surface roughness characterized by SEM. Images of cross-sections of PbLT-fabricated microfluidic chips (**a**) without hydrophobic treatment and (**b**) with hydrophobic treatment. The dashed line indicated by arrows on (**a**,**b**) shows the boundary between the PMMA substrate and Parafilm^®^. (**c**) Microchannel without hydrophobic treatment after infusion of PS beads. Obvious clusters were observed on the channel surface, especially at the corners. (**d**) The adhesion of 10 μm PS beads to the channel surface was significantly reduced by hydrophobic treatment.

**Figure 7 micromachines-14-00656-f007:**
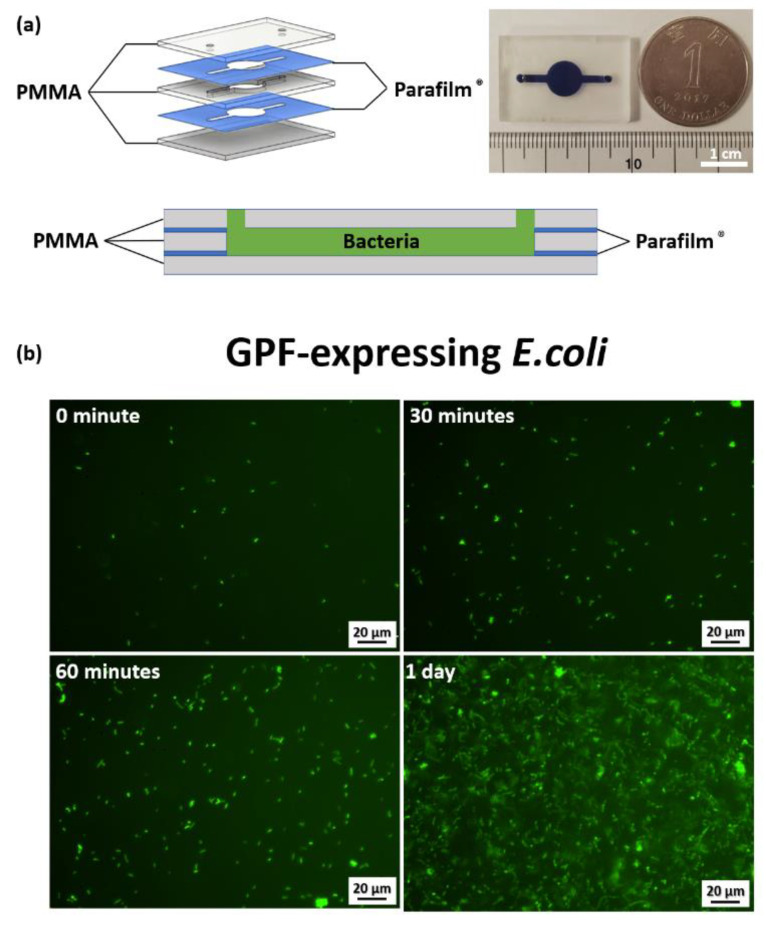
Bacteria cultivation in PbLT-fabricated bioreactor. (**a**) An exploded view and cross-sectional view of the bioreactor and the fabricated bioreactor. (**b**) Fluorescence images were acquired at 0, 30, 60 min, and 1 day post-cultivation in the bioreactor. (Scale bar = 20 μm).

**Table 1 micromachines-14-00656-t001:** Comparison of PbLT and conventional microfluidic chip fabrication methods.

	Soft Lithography	Injection Molding	Our Work
Set-up cost	~$80 k	>$50 k	<$10 k
Cost per print/materials	High	Low	Low
Cycle time	~24 h	3 weeks	<2 h
Throughput	Low	Very high	High
Difficulty in customized fabrication	High	High	Low

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
