# Peer review of "Rapid Prototyping of Multi-Functional and Biocompatible Parafilm®-Based Microfluidic Devices by Laser Ablation and Thermal Bonding"

_micromachines, 2023, doi:10.3390/mi14030656_

Round 1

Reviewer 1 Report

 The authors present the fabrication and exhaustive characterization of Parafilm-based microfluidic devices for different applications. The microfluidics were fabricated by laser ablation and thermal bonding of the substrates and parafilm layers. The article is clear and well written, however, there is no novelty. The reported approach is very similar to the work published by Lu et al. in 2016 (Fast prototyping of a customized microfluidic device in a non-clean-room setting by cutting and laminating Parafilm®), where the devices were fabricated by using a cutter plotter, to define the substrates and the parafilm layers, and a thermal bonding. Actually, by using a cutter plotter, many of the problems arisen by the heat produced during the ablation process may be solved.

I miss some description in the introduction of previous approaches in parafilm-based microfluidics:

https://doi.org/10.1039/C6RA18988A 2016 Fast prototyping of a customized microfluidic device in a non-clean-room setting by cutting and laminating Parafilm®

https://doi.org/10.1039/C5LC00044K   2015 Microfluidic paper-based analytical devices fabricated by low-cost photolithography and embossing of Parafilm®†

 https://doi.org/10.1021/ed500401d  Cost Effective Paper-Based Colorimetric Microfluidic Devices and Mobile Phone Camera Readers for the Classroom

 https://doi.org/10.1016/j.snb.2017.10.005    2018, Laminated and infused Parafilm® – paper for paper-based analytical devices

 I also miss a discussion/comparison of this previous works with the work presented by the authors.

Actually, the work from Lu et  al. is referenced in the article (ref. 15), as a reference for the text “The poisonous 366 chemicals (such as HF and SU8 solvent) are also contrary to the original goals of micro-367 fluidics development: inexpensive, convenient, portable, environmentally friendly, and 368 disposability”, which have no link with the article. However, it is not included/discussed in the introduction, which I found quite worrying.

Author Response

We thank the reviewer’s comments. Please see the point-by point response from the attachment.

Reviewer 2 Report

This paper describes a useful new technique for quick and cost effective microfluidic fabrication of multilayer devices. It will be useful for many researchers looking for rapid prototyping of microfluidic devices for a wide range of applications especially those in low resource settings.

However, the benefits are overstated, some statements are misleading and a direct comparison between other bonding methods is not made. I would recommend the following changes before the manuscript could be published:

Major:

It is essential suggest that the authors perform the same bonding strength tests on typical PDMS plasma bonding, or some other standard, so there is an actual comparison under the same conditions. I am extremely surprised by the statement “all 85.0 °C fabricated chips (18.8 MPa for 1 cm2) have shown larger bonding strength compared to PDMS-PDMS bonding (~1.9 MPa for 0.5 cm2) and PDMS-glass bonding (~3.3 MPa for 0.5). Given that PDMS-PDMS bonding can achieve covalent bonding I find it surprising that this method is so much stronger. Making a comparison between the authors experimental results and that in another paper may be hampered by differences in measurement techniques and experimental setup. A comparison under the same setup is needed.

Minor:

Please measure the height of the channels (the thickness of parafilm is stated but presumable once heated and compressed this may change)

Is the resolution limited by the optimal channel width, or the laser head? (i.e. does the laser cutter have a higher precision than 250um?) This will be important for those creating chips that are more complex than a straight line.

Figure 1. a star symbol is added above certain bars in the bar graphs, yet no statistical information is provided. State the stats used and P value, and which group is being compared with which.

“In tensile testing, Parafilm® has much larger ultimate tensile strengths(>2 MPa) compared to PDMS (< 1 MPa).”  According the source that the authors cite for the tensile strength of PDMS this is not correct. The source sates “young's modulus of 1.32–2.97 MPa, ultimate tensile strength of 3.51–7.65 MPa”. Thus PDMS has a higher tensile strength not parafilm.

Line 242  “Overall, compared to PDMS, Parafilm® has better elasticity and resistance to tensile.” Should be revised accordingly

Line 264: “Compressed nitrogen gas (maximum 0.2 MPa) was injected into fabricated chips for gas leakage tests. For the chips fabricated at 85.0 °C, minimal leakage was observed.” Where is this quantified. Minimal leakage suggests there was some. At what pressure did it leak?

Figure 7. The longer term images of bacteria should be shown too. Can a growth curve be plotted?

“Noted that minimal leakage was observed during the 7 days of bacterial cultivation” Please be more clear, some devices did leak? Or all leaked slightly? This is important, other bonding techniques produce irreversible bonding that will not leak over long term culture.

Table 1. is misleading and slightly subjective:

In “3D compatibility” injection moulding and softlithography are stated as having Layered 2D designs Layered 2D designs while the authors approach is suggested as being better, sated as “3D designs”. However the 3D approach is just layered 2D designs and not true 3D as can be achieved by 3D printing. The authors should change this to “Layered 2D”.

fluid automation”. I presume the authors are referring to integration of valves? If so please change to this. And if so, the authors technique has a significant drawback, which they state “It has been observed that for the type 1 chip, the flow rate decreased from over 20.0 μL/s to 0 μL/s within 25s, which presumably dues to the fatigue failure of Parafilm®” PDMS passed valves are much more robust.

Manufacturability is a very loose and not defined term. I would argue that injection moulding by far is the most apt for large scale manufacturing out of all these techniques. It is not clear why the authors have rated their technique so highly in this category.

Supplementary table 1 the burst pressure of PDMS to glass is given at 0.1, since the cited article gives it as 0.18, I recommend rounding up to 0.2 or giving the precise figure of 0.18. However other sources cite up to 0.27MPa. https://www.ncbi.nlm.nih.gov/pmc/articles/PMC8394141/

 A range of values would be most appropriate

Author Response

(The authors gave the same response as above.)

Reviewer 3 Report

This paper mentions microfluidic channel fabrication using laser ablation and thermal bonding of Parafilm. This technique may be interesting for potential readers, and suitable for publication in micromachines. However, the reviewer recommends to revise the present manuscript for improvement of readability. Therefore, this paper can be accepted for publication after reflecting following comments to the revised manuscript.

<Major comments>

#1: The authors must review previous studies. For example, thermal bonding methods for microfluidic channel preparation and micropump using elastomers such as PDMS in introduction.

#2: Add dimensions of the parafilm before and after stretching, and the microfluidic devices. Especially, thickness of Parafilm M, microfluidic channel heights of Parafilm layers, and microfluidic channel dimensions of Figure 5 and 7 should be shown.

#3: The reviewer found some questions about Micropump section. What is the difference between type 1 micropump and type 1 chip in page 10? Why the pumping speed increased compared to PDMS micromembrane?

<Minor comments>

#1: Figure 2 and 3, The arrows “->” should be “” (use special character).

#2: Caption of Figure 2, (d) “laser output power of 10.0 W” is not consistent with the text inside the figure “Power: 12 W”. Which is correct?

#3: Figure 6, it is difficult for readers to recognize the microbeads in the SEM images. Can the authors improve the resolution of the images, if possible?

#4: Line 352, Erase “23” at the end of the sentence.

#5: Figure 7; Can the authors count the bacterial cells in the pictures of Figure 7(b)?

#6: Recheck format of ‘References’ section.

Author Response

(The authors gave the same response as above.)

Round 2

Reviewer 1 Report

The authors have made changes according to reviewers suggestions and have clarify some of the critical aspects related with the novelty.

Reviewer 2 Report

I would like to thank the authors for their response. They have made a number of improvements. However please address the following issues before publication:

In response to comment 1 the authors provide some new experimental data in which they compare bond strength between their method and plasma PDMS bonding. They conclude that their method is superior however in their data it seems that there is a large variation in peak load using their method. The authors should do a statistical test to see if there is a significant difference between methods for both shear and peak load. The authors should also comment on the high level of variation in peak load seen in their devices.

In response to Comment #9 the authors have provided new data showing devices that leaked. This should be provided in the supplemental data, and a clear statement should be made in the main text as to the number of devices that leaked.
